# Diversity and Genetic Structure of *Theileria annulata* in Pakistan and Other Endemic Sites

**DOI:** 10.3390/pathogens11030334

**Published:** 2022-03-10

**Authors:** Salama Al-Hamidhi, Asia Parveen, Furhan Iqbal, Muhammad Asif, Naheed Akhtar, Elshafie I. Elshafie, Albano Beja-Pereira, Hamza A. Babiker

**Affiliations:** 1Department of Biochemistry, College of Medicine and Health Sciences, Sultan Qaboos University, Muscat 123, Oman; salama.2012@gmail.com; 2Institute of Pure and Applied Biology, Zoology Division, Bahauddin Zakariya University, Multan 60800, Pakistan; asia@hotmail.com (A.P.); furhan.iqbal@bzu.edu.pk (F.I.); asifm@hotamil.com (M.A.); akhtarn@hotmail.com (N.A.); 3Department of Animal and Veterinary Sciences, College of Agricultural and Marine Sciences, Sultan Qaboos University, Muscat 123, Oman; elshafiei@squ.edu.om; 4Central Veterinary Research Laboratories, Al Amarat, Khartoum P.O. Box 8067, Sudan; 5Research Centre in Biodiversity and Genetic Resources (CIBIO), InBIO, University of Porto, Rua Padre Armando Quintas 7, 4485-661 Vairão, Portugal; albanobp@gmail.com; 6DGAOT, Faculty of Sciences, Universidade do Porto, Rua Campo Alegre 687, 4169-007 Porto, Portugal; 7Sustainable Agrifood Production Research Centre (GreenUPorto), Universidade do Porto, Rua da Agrária 747, 4485-646 Vairão, Portugal; 8Institute of Immunology and Infection Research, School of Biological Sciences, University of Edinburgh, Edinburgh EH8 9YL, UK

**Keywords:** *Theileria annulata*, heterozygosity, genotype diversity, sub-structuring, Pakistan

## Abstract

Background: *Theileria annulata* is a tick-borne protozoan parasite responsible for bovine theileriosis, a disease that impacts cattle population in many developing countries. Development and deployment of effective control strategies, based on vaccine or therapy, should consider the extent of diversity of the parasite and its population structure in different endemic areas. In this study, we examined *T. annulata* in Pakistan and carried out a comparative analysis with similar data garneted in other areas, to provide further information on the level of parasite diversity and parasite genetic structure in different endemic areas. Methods: The present study examined a set of 10 microsatellites/minisatellites and analyzed the genetic structure of *T. annulata* in cattle breeds from Pakistan (Indian sub-continent) and compared these with those in Oman (Middle East), Tunisia (Africa), and Turkey (Europe). Result: A high level of genetic diversity was observed among *T. annulata* detected in cattle from Pakistan, comparable to that in Oman, Tunisia, and Turkey. The genotypes of *T. annulata* in these four countries form genetically distinct groups that are geographically sub-structured. The *T. annulata* population in Oman overlapped with that in the Indian Subcontinent (Pakistan) and that in Africa (Tunisia). Conclusions: The *T. annulata* parasite in Pakistan is highly diverse, and genetically differentiated. This pattern accords well and complements that seen among *T. annulata* representing the global endemic site. The parasite population in the Arabian Peninsula overlapped with that in the Indian-Subcontinent (India) and that in Africa (Tunisia), which shared some genotypes with that in the Near East and Europe (Turkey). This suggests some level of parasite gene flow, indicative of limited movement between neighboring countries.

## 1. Introduction

Protozoan parasites belonging to the *Theileria* genus infect a wide range of domestic and wild animals and are transmitted by Ixodid ticks of several genera including *Amblyomma*, *Haemaphysalis*, *Hyalomma*, and *Rhipicephalus* [1,2]. A number of *Theileria* species cause diseases that are a threat to the agricultural industry of many countries in sub-Saharan Africa and Asia, causing high animal mortality and reduced animal productivity and constraining livestock improvement [3]. The most economically important *Theileria* parasite species of cattle are *T. parva* and *T. annulata*. The form that infects cattle (sporozoite) induces the transformation of infected leukocytes, forming the schizont stage, resulting in lymphoproliferative diseases, East Coast fever, and tropical theileriosis [4]. The schizonts undergo further differentiation to merozoites; once released upon the lysis of infected cells, merozoites then enter erythrocytes, where they develop into the vector infective form known as piroplasms Forsyth, et al., 1997 [3]. Analysis of *T. annulata* in many endemic settings has revealed a high level of genetic diversity of the haploid parasite in infected cattle [5,6,7], consistent with a high frequency of recombination during the brief diploid phase of the life cycle in the tick vectors. This feature is also commonly observed in other related apicomplexan parasites like *Plasmodium* [8]. Thus, the high level of multi-locus genotype diversity seen in infected cattle and the lack of clustered genotypes led to the hypothesis of a panmictic population genetic structure of *T. annulata* [6]. However, isolation imposed by restricted host movement and geographic distance can lead to genetic differentiation of the parasite in different regions [8,9,10]. Variation has been seen between *T. annulata* populations among different regions, which, over time, can lead to the genetic divergence of the parasite populations [7]. Similar genetic differences have been reported for *T. parva* [9] and *T. lestoquardi* [10].

However, a low level of genetic differentiation among populations is often interpreted as an indicator of high dispersal rates and high effective population sizes in sampling locations [11,12]. A high and significant coefficient of correlation indicates spatial similarity between the genetic structures of the host and parasite populations, suggesting that host and parasite dispersal are strongly related to each other [13]. Such a correlation is likely to exist for *Theileria* spp. as the movement of these parasites and their tick vectors are thought to be mediated by the extensive anthropogenic movement of the livestock, given that tick migration is limited [14,15,16].

The present study added to the above findings and examined the genetic diversity of natural *T. annulata* populations in cattle from Pakistan, where bovine theileriosis is a major livestock constrain [17]. To allow cooperative analysis, the study employed 10 microsatellites, distributed over the four parasite chromosomes [6], that have been used for analysis of the parasite in different countries [5,6]. The data, obtained from the *T. annulata* in Pakistan, was compared to similar data originating from Oman, Tunisia, and Turkey to assess the level of genetic differentiation of *T. annulata* populations in geographically and climatically distinct regions.

## 2. Results

### 2.1. Demographic Data

A total of 841 cattle, representing two breeds, Sahiwal (n = 299), Friesian (n = 242), and a crossbred population (n = 300), were screened for the presence of *T. annulata* by PCR assay targeting the *18S* ribosomal gene of the parasite. The majority of cattle (n = 617; 73.4%) were infested with adult ticks and carried *T. annulata* (n *=* 133; 15.82%).

### 2.2. Diversity of Microsatellites and Minisatellites

Eighty-two *T. annulata* isolates from Pakistan were successfully genotyped for 10 *T. annulata*-specific microsatellites and minisatellites. A high level of polymorphism was observed among the examined *T. annulata* isolates, with the total number of alleles for each marker in the population ranging from six for Ts25 to 29 for Ts8. The average heterozygosity was high (*H**_e_* = 0.743), ranging from 0.348 to 0.960 (Table 1). This is comparable to that reported for the parasite populations in Oman (*H_e_* = 0.842) [18], Tunisia (*H_e_* = 0.866) [7], Turkey (*H_e_* = 0.865) [7], and India (*He* = 0.864) [6], measured using the same panel of markers (Table 1).

### 2.3. Population Differentiation

A higher level of *F_ST_* values was detected between each pair of the four *T. annulata* populations (Table 2). The highest *F_ST_* was seen between parasite populations in Pakistan and Turkey (*F_ST_* = 0.166), followed by Pakistan and Tunisia (*F_ST_* = 0.129). The next highest level of differentiation was seen between Oman and Turkey (*F_ST_* = 0.109) as well as Pakistan and Oman (*F_ST_* = 0.096). However, a moderate genetic differentiation was shown between Oman and Tunisia (*F_ST_* = 0.063) and Tunisia and Turkey (*F_ST_* = 0.052) (Table 2).

We tested the effect of geographical separation on population differentiation, Nei’s genetic distance (*D*) (Nei, 1978) was estimated between populations from Pakistan, Oman, Tunisia, and Turkey. Regression analysis showed no correlation between pairwise *F_ST_* values and the geographical distance between *T. annulata* populations from Pakistan, Oman, Turkey, and Tunisia (*R^2^* = 0.0704, *p* = 0.3) (Table 2 and Figure 1A). There was a trend for differentiation by distance, but this was not significant.

The above levels of differentiation are also depicted by PCoA analysis. A discrete clustering of parasites from Turkey and Pakistan was evident, indicating that geographical separation is probably associated with genetic isolation of the parasite populations (Figure 1B). However, there was an overlap between the genetic diversity of *T. annulata* from Oman and Tunisia. Similarly, the *T. annulata* population from Oman displayed overlapping with that in Pakistan. Structure analysis clearly indicated the presence of four discrete clusters representing underlying sub-populations (K = 4). MLGs of each of the four populations formed a discrete cluster, with limited overlap between parasites from Oman and Pakistan and between parasites from Oman and Tunisia (Figure 1B and Figure 2).

## 3. Discussion

The study described the genetic structured of *T. annulata* in Pakistan and compared the data to that reported in other endemic areas in the Middle East, Africa, and Europe.

A large amount of genetic diversity was observed among *T. annulata* isolated from cattle in Pakistan, comparable to those isolated from Oman, Tunisia, and Turkey. The above diversity is probably driven by the multiplicity of *T. annulata* genotypes in an infected host and sustained transmission by the tick vector, despite variability of their species in the above regions. Genotype multiplicity in infected cattle could sustain a high rate of cross-mating and recombination in atick vector, which, in turn, would result in increased genetic diversity in the bovine host, as demonstrated for the human malaria parasite *P. falciparum* [7].

All the animals included in the current study were apparently healthy with no clinical signs of theileriosis. It is, therefore, possible that these animals were carrying a persisting *T. annulata* infection. In a previous cohort study in Oman, we found that *Theileria lestoquardi* can persist as asymptomatic in infected sheep for a period of over 13 months. During this period, some genotypes persist and remain stable over time, while others fluctuated regularly within an infection (24). However, in the current study, we did not detect multiple genotype multiplicity with infected cattle. Therefore, analysis of repeated samples from individual infection would not change the conclusion drawn.

The parasites in these four countries tend to form genetically distinct groups with some geographical sub-structuring (Figure 1B and Figure 2). The relatively lower level of genetic differentiation detected between *T. annulata* from Pakistan and Oman (*F_ST_* = 0.09) when compared with that observed between Pakistan and Tunisia (*F_ST_* = 0.1293) and Pakistan and Turkey (*F_ST_* = 0.166), is consistent with the pattern of a high level of differentiation documented between samples of *T. annulata* from Oman and Tunisia (*F_ST_* = 0.06855) and from Oman and Turkey (*F_ST_* = 0.1092) [20]. Similar observations regarding the genetic diversity of *T. annulata* have been reported in a recent study from India by Roy et al. [6]. Moderate to high genetic differentiation (*F_ST_* = 0.0573 to 0.085) was observed when the Indian *T. annulata* population was compared with that in Tunisia and Turkey [6]. However, analyses of *T. annulata and the reacted species*, *T. lestoquardi*, from distinct locations in a single country either show no or limited genetic differentiation [6,21,22]. Therefore, while the current study analyzed *T. annulata* in a single region in Pakistan, the genetic profile of the parasite in this region is not expected to be distinct compared to those that exist in other regions in the country.

The above genetic differentiation of the parasite populations did not correlate with the geographical distances. These results contrast with other studies that demonstrated a correlation between pairwise FST values and the geographical distance between *T. annulata* in widely separated populations, where gene flow is hindered by geographical and trade barriers [5,7]. However, the current study is based on relatively small samples of *T. annulata* populations in Pakistan compared to data generated in Oman and Tunisia (Table 2). Thus, the results must be interpreted with caution and should be viewed as preliminary evidence for lack of a clear differentiation between geographically separated populations. Further studies using a much larger number of *T. annulata* isolates collected from cattle and ticks in different regions in Pakistan would be necessary to further test the hypothesis of geographical sub-structuring of *T. annulata* populations.

Despite the lack of association between genetic and geographical distances among *T. annulata* populations, the PcoA depicts a visible separation between Pakistan and Oman relative to Tunisia and Turkey. Nonetheless, there was some overlap between parasites in Tunisia and Oman, indicating that the separation between the parasite populations in the two countries was incomplete (Figure 1B). Similar incomplete separation has been previously reported in *T. annulata* populations from widely separated countries [7,20]. These observations suggest that despite ecological and epidemiological barriers between the different parasite populations, gene flow occurs between the above countries, probably due to the movement of animals, often as an asymptomatic infection and infesting ticks. [23,24].

In addition to host movement, many studies have demonstrated the role of migratory birds in the dispersal of ticks and tick-borne pathogens, including several potential zoonotic pathogens [25]. There is currently no information on ticks in migratory birds in Pakistan and the extent of imported *Theileria* spp. via this potential route. However, some studies have demonstrated the role of migratory birds in the dissemination of Avian influenza viruses into domestic poultry through migratory wild birds from Pakistan, in sites situated across the migratory Indus flyway [26]. With regard to tick dispersal, detection of *Hyalomma* spp. in Central Europe (Germany) outside their distribution zones in tropical and sub-tropical areas has been linked to the western migratory route of birds from West Africa [27]. All *Hyalomma* specimens detected in Germany were adult ticks sampled in large animals. However, nymphal ticks were collected from migrating birds in Scandinavia [28]. This suggests that tick species are molting from the nymphal stage to the adult stage outside the usual distribution area [29]. Therefore, it can be assumed that every year a large number of immature *Hyalomma* ticks and some associated pathogens are transported via migratory birds outside its origin sites [30]. The above emphasize the role of imported *Hyalomma* ticks, via migratory birds, in the introduction of novel variants in endemics areas in the seasonal migratory routes. This process has also been implicated in the high diversity seen in other apicomplexan parasites, such as *Plasmodium falciparum*. Many studies have concluded that extensive human movement has led to a lack of geographical sub-structures of *P. falciparum* across many countries in sub-tropical Africa [31,32]

Furthermore, one should not exclude the possible role variation in the distribution of Ixodes tick species on the extent of the diversity of *T. annulata* over the study regions. The primary vector for *T. annulata* is *Hyalomma* spp. (3), with multiple species distributed across a wide geographical spectrum (Southern Europe, North Africa, East Asia). While *H. anatolicume* is the main vector in the study sites, many additional species of *Hyalomma* exist in some [18,20,22]. The variation in the intrinsic characteristics of these ticks, such as feeing behavior, can influence their transmissibility capacity and the extent of diversity of *T. annulata* over the study regions.

Therefore, infected host movement and possible tick dispersal via migratory birds can explain the similar level of diversity of *T. annulata* from Oman with that from Africa and the Indian Subcontinent as cattle trade is common between these regions [33]. An effective control and elimination strategy of bovine Theileriosis requires better understanding of *T. annulata* in the region, in particular the origin and spread of putative drug- and vaccine-resistant strains. A regional population genetic network could help to improve knowledge on genetic relatedness and routes of movement. It would, therefore, be of interest to examine *T. annulata* populations in neighboring countries in the region, where theileriosis is also a major problem to estimate the genetic relationships among populations. By doing this, it would be possible to determine whether control measures, based on vaccine or drug therapy, should be implemented separately at a national level or if a regional policy can and should be adopted. In addition, policies to restrict the influx of putative drug- and vaccine-resistant strains can be imposed.

In summary, the present study revealed a high level of genetic diversity of Pakistani *T. annulata*, where the parasite is genetically differentiated from other populations of geographically distant endemic countries. Finally, the closer genetic relatedness observed between *T. annulata* from Pakistan and Oman compared with *T. annulata* from Tunisia and Turkey provides evidence that the populations from these latter two countries are not connected and are evolving differentially.

## 4. Materials and Methods

### 4.1. Study Site and Sample Collection

A total of 844 blood samples were collected during 2017 and 2018, from apparently healthy (Holstein Friesian, Sahiwal, and cross breed) cattle in the Layyah District of Southern Punjab (Pakistan) (Figure 3). A previous molecular survey in this region found that crossbreed cattle were the most susceptible to *T. annulata* (28%), followed by Sahiwal (19%) and Holstein (14%). Layyah District lies between 30°45′ to 31°24′ N and 70°44′ to 71°50′ E. The area consists of a semi-rectangular block of sandy land between the Indus River and the Chenab River in Sindh Sagar Doab. The area covered by the district is 6291 km^2^ with a width from east to west of 88 Km and a length from north to south of 72 km. In Layyah, the summers are long, sweltering, humid, and clear; the winters are short, cool, and mostly clear and it is dry year-round. Over the course of the year, the temperature typically varies from 7 °C to 41.1 °C and is rarely below 4.4 °C or above 45 °C [2].

After obtaining informed consent from the owners, the animals included in this study were examined by a professional veterinarian and blood samples (each approximately 10 mL) were collected in EDTA tubes from the jugular vein for DNA extraction.

### 4.2. Identification of Theileria Species and Microsatellite and Minisatellite Genotyping

DNA was extracted using a Qiagen QIAamp DNA mini Kit (Qiagen, Germany) following the manufacturer’s instructions and was stored at −20 °C. *Theileria* genus-positive samples were confirmed by polymerase chain reaction (PCR) using pan-*Theileria* primers [forward 5′-GGC GTT TAT TAG ACC TAA AAC CAA AC-3′ and reverse 5′-TTT GAG CAC TCT AAT CTC AAA GT-3′], targeting the *18S ribosomal gene* (rDNA) as described elsewhere [34]. PCR was followed by RFLP analysis of all positive samples to discriminate different *Theileria* species. Known DNA samples of *T. annulata*, *T. ovis*, and *T. lestoquardi* were included as control for the assay.

A panel of 10 polymorphic microsatellite and minisatellite markers designed for the genetic analysis of *T. annulata* was used to genotype all positive samples included in the present study. Details of the method and primers used are the same as those described previously [7,21]. Briefly, the 5′ end forward PCR primer in each locus was labeled with a fluorescent dye (FAM or HEX). PCR was carried out in a total volume of 25 µL containing 2 µL of template DNA, 1 µL of each primer (10 pmol), 1U of Taq polymerase (Biolab, UK), and 10 mM dNTPs. The PCR protocol consisted of an initial denaturation at 95 °C for 5 min followed by 32 cycles of denaturation at 95 °C for 30 s, annealing between 42 to 62 °C (specific for each marker) for 30 s, extension at 65 °C for 30 s and an overall extension at 65 °C for 5 min. Capillary electrophoresis was carried out in the Genetic Analyser ABI3130 xl (Applied Biosystems, Warrington, Cheshire, UK) to determine the sizes of PCR products relative to the size-standards ROX-labeled GS500 (Applied Biosystems) using Genemapper software (Applied Biosystems, UK).

The predominant allele at each locus was identified to generate a multi-locus genotype (MLG) that represents an estimate of the most abundant genotype in each sample. The MLG data was then used to assess population genetics parameters such as genetic diversity, gametic linkage disequilibrium, genetic structure, and population differentiation.

### 4.3. Genetic Analysis

The MLG dataset was compared to previously generated datasets from *T. annulata* isolates in Oman [5], Turkey, and Tunisia [7] in order to determine the extent of diversity and genetic relatedness among the four populations.

Genetic diversity parameters, including an estimation of the number of alleles per locus and expected heterozygosity *(H_e_*) were calculated for the entire dataset using GenAlex v6.5 [35] in order to assess the polymorphism level at each locus and to determine the overall diversity as well as the diversity within the sub-populations. The expected heterozygosity was calculated using the formula for ‘unbiased heterozygosity’, also termed ‘haploid genetic diversity’, *H_e_* = [*n*/(*n* − 1)][1 − ∑*p*^2^] where *n* is the number of isolates and *p* is the frequency of each different allele at a locus [36].

To determine the genetic structure of *T. annulata* populations, whether they comprised of a single panmictic population with a high degree of genetic exchange or not, the multi-locus linkage disequilibrium (LD) of the alleles at pairs of loci was measured using the standard index of association (I^S^_A_). Both I^S^_A_ and the variance of data were calculated using the program LIAN version 3.5 [37].

Population differentiation was assessed by estimating Wright’s *F_ST_* index using the Fstat computer package Version 2.9.3.2. Two estimators of *F_ST_* (G′_ST_ and θ) [38,39] were used to estimate genetic differentiation between sub-populations. Principal Co-ordinate Analysis (PCoA) was performed using GeneAlex6 to visualize the relationships between MLGs. In addition, a test for isolation by distance was carried out using the pairwise genetic distances (*F_ST_*) and geographical distances (km) between populations, using the Mantel test of the correlation between the two distance matrices (GenAlex v6.5) [30].

Finally, the software Structure V. 2.3.4 was used to deduce the underlying structure of parasite population and to determine the number of its sub-populations. This method detects clusters without prior information on the origin of samples. Ten iterations for K = 2 to K = 6 (K being the number of clusters) were run, each with a burn-in period of 10,000 steps and then 20,000 Markov Chain Monte Carlo (MCMC) iterations [40]. In addition, the optimal number of clusters was assessed using the program structure [41].

## Figures and Tables

**Figure 1 pathogens-11-00334-f001:**
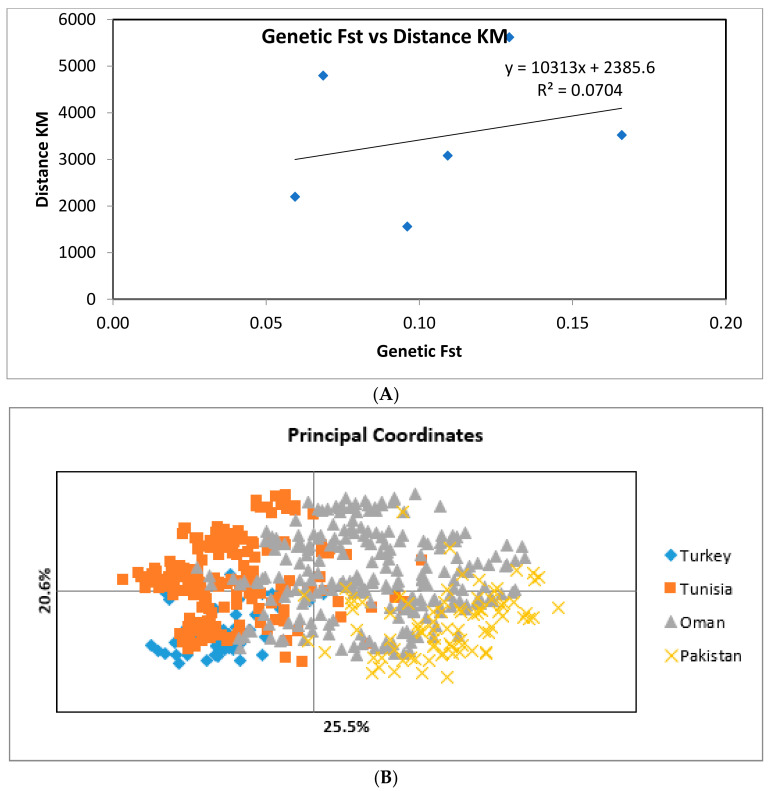
(**A**) Pairwise *F_ST_* estimates and geographical distance (in Km) between *T. annulata* populations in Pakistan, Oman (North & South Oman) Tunisia, and Turkey. (**B**) Principal Co-ordinate Analysis (PCoA) of *T. annulata* populations from Pakistan, Oman, Turkey, and Tunisia. The amount of variation in the dataset represented by each axis is presented as a percentage.

**Figure 2 pathogens-11-00334-f002:**
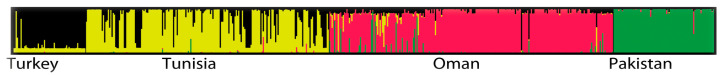
‘Structure’ analysis of *T. annulata* genotypes from Pakistan, Oman, Tunisia, and Turkey. The presence of four sub-populations was inferred (K = 4), K4 (Delta K = 1.9). The population structure and the Evanno approach were used to determine K [19]. The bar plot illustrates the population structure at K = 4 in *T. annulata* in Pakistan and parasites from Oman, Tunisia, and Turkey. Each vertical bar represents an individual sample, and each color represents one of the K clusters (subpopulations). Delta K indicates that K = 4.

**Figure 3 pathogens-11-00334-f003:**
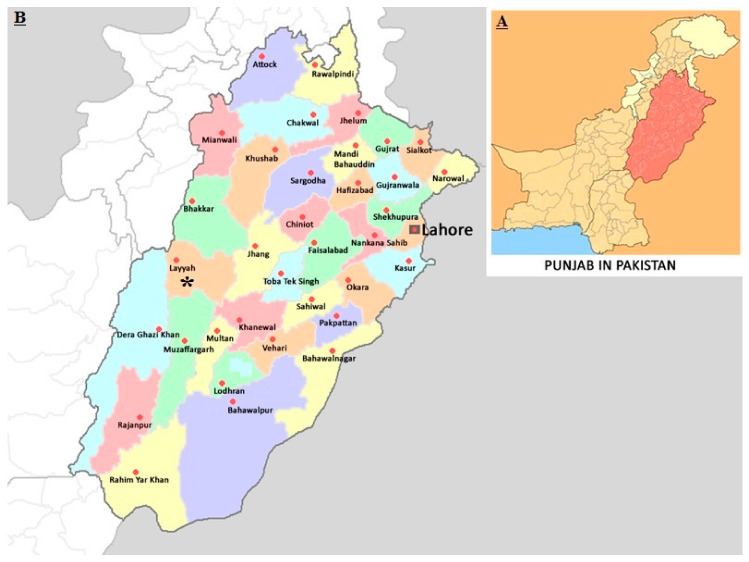
(**A**) Map of Pakistan with highlighted Punjab province. (**B**) District Layyah, where cattle blood samples were collected from, is marked with * in magnified section.

**Table 1 pathogens-11-00334-t001:** Allelic diversity and unbiased heterozygosity (He) at 10 microsatellites and minisatellites loci among *T.*
*annulata* isolates in Pakistan, Oman, Turkey, and Tunisia.

	n	TS5	TS6	TS8	TS9	TS12	TS15	TS16	TS20	TS25	TS31	Average *He*
Pakistan	82	0.627	0.941	0.960	0.922	0.935	0.638	0.613	0.743	0.348	0.704	0.743
Oman	231	0.781	0.887	0.937	0.916	0.945	0.862	0.735	0.667	0.769	0.920	0.842
Turkey	59	0.827	0.941	0.953	0.931	0.936	0.829	0.828	0.897	0.669	0.836	0.865
Tunisia	198	0.826	0.943	0.957	0.900	0.957	0.815	0.803	0.851	0.658	0.950	0.866

**Table 2 pathogens-11-00334-t002:** Pairwise *F_ST_* estimates and geographical distance (in Km) between *T. annulata* populations in Pakistan, Oman, Tunisia, and Turkey.

	Turkey	Tunisia	Oman	Pakistan
Turkey	0	2200.9	3083.35	3526.09
Tunisia	0.0594	0	4800.6	5621.92
Oman	0.1092	0.06855	0	1562.12
Pakistan	0.166	0.1293	0.096	0

## Data Availability

The authors confirm that the data supporting the findings of this study are available within the article if required.

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
