# Peer review of "Diversity and Genetic Structure of Theileria annulata in Pakistan and Other Endemic Sites"

_pathogens, 2022, doi:10.3390/pathogens11030334_

Round 1

Reviewer 1 Report

Authors did a great job in the revision process. The quality of the text has improved considerably. I have minor corrections to be made.

Results:

Line 84: Change 844 to 841 cattle

Reconsider the sentence between lines 107 to 11, since the coefficient of determination had a low value. This shows that the equation generated in linear regression has a low predictive power. Thus, there is no relationship between the genetic fst and the distance between populations.

R2 value in text is different from figure 1A. 

Discussion:

Remove the sentence "The presents study reiterated the impact of bovine theileriosis and demonstrates hematological changes associated with T. annulata infection in cattle in Pakistan". The hematological analysis was excluded from the work. Rewrite this paragraph. 

Delete "However" in line 149.

Remove from the discussion the text referring to the influence of distance on the genetic variability of Theileria annulata. Authors cannot make this claim in the paper. 

Author Response

Reviewer 1

Comments and Suggestions for Authors

Authors did a great job in the revision process. The quality of the text has improved considerably. I have minor corrections to be made.

Response: We thank the reviewer for the favorable comments on the manuscript

Results:

  1. Line 84: Change 844 to 841 cattle

Response: Done

  1. Reconsider the sentence between lines 107 to 11, since the coefficient of determination had a low value. This shows that the equation generated in linear regression has a low predictive power. Thus, there is no relationship between the genetic FST and the distance between populations.

Response: The above section has been modified to reflect the lack of correlation between FST and geographical distance

  1. R2 value in text is different from figure 1A. 

Response: We thank the reviewer for this has been corrected

Discussion:

  1. Remove the sentence "The presents study reiterated the impact of bovine theileriosis and demonstrates hematological changes associated with annulata infection in cattle in Pakistan". The hematological analysis was excluded from the work. Rewrite this paragraph. 

Response: The above sentence has been removed and the paragraph has been edited.

  1. Delete "However" in line 149.

Response: Done

  1. Remove from the discussion the text referring to the influence of distance on the genetic variability of Theileria annulata. Authors cannot make this claim in the paper. 

Response: this section has been edited to clarify our results that showed a lack of correlation between the pairwise FSTand geographical distance. In addition, extra sentences were added to interpret these results in context of T. annulatapopulation differentiation demonstrated by FST.

Reviewer 2 Report

In this study, the authors utilize whole blood samples from cattle in one province in Pakistan to determine genetic diversity of Theileria annulata isolates via analysis of mini- and micro-stellite markers.  They then compare these isolates to those in selected other  countries. 

I have the following comments and questions:

  1. The authors must perform extensive English language and grammatical editing.
  2. Please explain what assays were used to determine whether animals were positive for T. annulata.  Were the animals clinically ill? Had they received any treatment? 
  3. A major limitation of this study is the limited geographic s ope of the sample pool.   Inclusion of samples from cattle in other regions would greatly strengthen the results.
  4. What breeds of cattle (and number of each) actually comprised the smaller, T. annulata group?
  5. Were the animals acutely or persistently infected? Is there variation in the ability of T. annulata strains to persist? Might this impact your findings? Please discuss.
  6. Please clarify how this information might be used to control T. annulata.

Author Response

Reviewer 2.

In this study, the authors utilize whole blood samples from cattle in one province in Pakistan to determine genetic diversity of T. annulata isolates via analysis of mini- and micro-satellite markers.  They then compare these isolates to those in selected other countries. 

I have the following comments and questions:

  1. The authors must perform extensive English language and grammatical editing.

Response: The manuscript was edited for English language, grammar, spelling and consistency of style throughout

  1. Please explain what assays were used to determine whether animals were positive for annulata.  Were the animals clinically ill? Had they received any treatment? 

Response: All the animals enrolled in Pakistan were apparently healthy without any clinical signs of theileriosis and animals did not receive any treatment against theileriosis.

Regarding the assays for detection of Theileria sp., extra information was added, lines 234-238, in the Methods section describing the primers and PCR and other assays used for identification of T. annulata

  1. A major limitation of this study is the limited geographic scope of the sample pool.  Inclusion of samples from cattle in other regions would greatly strengthen the results.

Response. We agree with the reviewer that adding extra data, from other regions in Pakistan, would add extra information on whether parasites in different parts of the country are differentiated or if they comprise a single interbreeding population. 

Analysis of T. annulata in widely separated regions in other countries showed very low levels of genetic differentiation between T. lestoquardi populations (Ref 15). These data suggest a rate of genetic exchange and gene flow between parasites in different parts of the country, sufficient to allow the population to remain homogenous and to overcome genetic drift through geographical and genetic isolation.

The cattle included in this study were from three districts in Pakistan, Punjab Province: Dera Ghazi Khan, Layyah and Lodhran (Figure 1). The three districts are far from each other but still present in Punjab province and movement of tick infested animals can facilitate parasite migration and gene flow. 

  1. What breeds of cattle (and number of each) actually comprised the smaller, annulata group?

Response Three cattle breeds were enrolled (Sahiwal, Holstein Frisian and cross breed). A pervious molecular survey found that crossbred cattle was the most susceptible to T. annulata infection (28%) followed by Sahiwal (19%) and Holstein Frisian (14%) (Ref 1).

  1. Were the animals acutely or persistently infected? Is there variation in the ability of annulata strains to persist? Might this impact your findings? Please discuss.

Response. All the animals included in the current study were apparently healthy without any clinical signs of theileriosis. It is therefore, possible that these animals were carrying a persisting T. annulata infection. In a previous cohort study in Oman, we found that T. lestoquardi persist as asymptomatic in infected sheep for a period of over 13 months. During this period some genotypes persist and remain stable over time, while others fluctuated regularly within an infection (Ref 24).  However, in the current study we did not detect multiple genotype multiplicity with infected cattle. Therefore, analysis of repeated samples from individual infection would not change the convulsion drawn.

We added the above an extra short para (lines 151-159) to discuss to discuss asymptomatic T. annulata infection among examined cattle in Pakistan.

  1. Please clarify how this information might be used to control annulata.

Response: Extra information has been added, the last para of the discussion, before summary, lines 196-205, to highlight possible application of the findings of the present study, and other similar studies on population genetics to improve on control of bovine theileriosis.  

Reviewer 3 Report

This MS is a short note on genetic identity and variation of /Theileria annulata/ identified in bovine blood in Pakistan in comparison to similar data from Oman, Tunisia and Turkey. The intention of the authors was to assess genetic similarity of /T. annulata/ strains and compare these among geographically distant populations (eg. Pakistan to Turkey, Oman and Tunisia). While the idea itself is interesting, the authors oversimplified the study, with concluding that genetic distances are (1) in correlation with geographical distances, (2) this is caused by genetic make-up of primary hosts (cows). There are several problems with the authors’ approach, making the conclusions of the MS highly questionable.

 First of all, the main hypothesis: that host populations and their movements are solely the cause of diversity in /T. annulata/ genetic  stocks and these should show a linear correlation with the geographic distances of their locations. While movements of cows (due to international trade primarily) are high, with intermixing at important trade-hubs, this is not the sole cause of genetic diversity in their respective pathogens. It is highly unlikely that Friesian stocks of Pakistani cows introduced the /T. annulata/ genetic lineages to Pakistan (or any other site, by the way). /Theileria annulata/ is vectored by ticks (primarily /Hyalomma/ spp., but not exclusively). There are at least 10 different /Hyalomma/ spp. in Pakistan and similar numbers are present in Turkey, Tunisia and Oman, too. While there is a small overlap between these tick species among the listed countries, each region has several endemics, too (at least compared to the other regions). Thus, one should not exclude the cause of vector linked genetic diversity in case of so many different, locally present vector species. Another logical fallacy associated with this hypothesis is the fact that the authors suggests that bird-flight distances between different sites are corresponding to exact trade-routes of livestock. While geographic distances may be correlative in case of transhumance, this is definitely not the case when sea fare is involved (eg. between these countries) and trading hubs are introduced (as in real-world situation).

 One more issue is the mingling of local/regional data on haplotype diversity of parasites. The analysed hosts in each country used for comparison came from several distinct populations (eg. SW and Central Turkey, Northern and Central Tunis – without exact locations! –, NW, N, NE and SW Oman. While in the case of Oman, the authors treat distinctly two populations (North vs. South) in the case of the other two countries no distinction is made. This approach may introduce a significant geographical bias (the only measure of ‘host diversity’ in the case of this MS is geographical distance between host populations), as for example Aydin (W TR) is chiefly at the same distance to Tunis (TS) as N Oman to Punjab, although genetic distances between these two pairs differ with a magnitude. Moreover, the geographic distance between N and S Oman is small, still the genetic divergence is high. Thus, just geographic distance between host populations per se does not determine haplogroup diversity and may be is useless for such comparisons.

 The third important issue is the way authors link host-population connectivity to geographical distance. This is entirely wrong, as nowadays genetic diversity in livestock (and especially in commercially bred, globally distributed races in cows) is not maintained exclusively by livestock exchanges as in the past, but by using seminal fluids of selected bulls. And most breeders can and do select from the widest palette of genetic strains, without any correspondence to geographical source origin…

 And a comment on the scope of the study: How will these data improve T. annulata control? The authors need to explain this either in the introduction or the discussion.

 There are several minor issues also:

 1. Language and grammar. Especially the Abstract and several paragraphs of the Discussion are hard to understand and bear many punctuation and grammatical errors.

 2.L33-34 Abstract/Conclusions: There is no such thing as ‘global endemic site’! It is either global or endemic… Moreover the whole sentence is meaningless.

 3.L71-72 Ticks does not migrate, however they are transported. By the livestock they’re attached and with migratory birds. This later is much more important and at least a magnitude larger, as involves millions of birds transiting this region. So, Hyalomma ticks definitely move between the mentioned areas independent of cattle trade.

Author Response

This MS is a short note on genetic identity and variation of Theileria annulata identified in bovine blood in Pakistan in comparison to similar data from Oman, Tunisia and Turkey. The intention of the authors was to assess genetic similarity of T. annulata strains and compare these among geographically distant populations (eg. Pakistan to Turkey, Oman and Tunisia). While the idea itself is interesting, the authors oversimplified the study, with concluding that genetic distances are (1) in correlation with geographical distances, (2) this is caused by genetic make-up of primary hosts (cows). There are several problems with the authors’ approach, making the conclusions of the MS highly questionable.

Response: We are grateful to the editor for the positive comments regarding the manuscript. We have restructured the manuscript and focused on genetic diversity of T. annulata strains in Pakistan, and compared these among geographically distant populations (eg. Pakistan to Turkey, Oman and Tunisia).

First of all, the main hypothesis: that host populations and their movements are solely the cause of diversity in T. annulata genetic stocks and these should show a linear correlation with the geographic distances of their locations. While movements of cows (due to international trade primarily) are high, with intermixing at important trade-hubs, this is not the sole cause of genetic diversity in their respective pathogens. It is highly unlikely that Friesian stocks of Pakistani cows introduced the T. annulata genetic lineages to Pakistan (or any other site, by the way). Theileria annulata is vectored by ticks (primarily Hyalomma spp., but not exclusively). There are at least 10 different Hyalomma spp. in Pakistan and similar numbers are present in Turkey, Tunisia and Oman, too. While there is a small overlap between these tick species among the listed countries, each region has several endemics, too (at least compared to the other regions). Thus, one should not exclude the cause of vector linked genetic diversity in case of so many different, locally present vectors. Another logical fallacy associated to this hypothesis is the fact that the authors suggests that bird-flight distances between different sites are corresponding to exact trade-routes of livestock trade. While geographic distances may be correlative in case of transhumance, this is definitely not the case when sea fare is involved (eg. between these countries) and trading hubs are introduced (as in real-world situation).

Response: We agree with the reviewer that the role of the host’s movement is not the solecause of dispersal and diversity of T. annulata genetic, and that the tick vector dynamics can play an important role in the parasite genetics. We have removed some parts of the manuscript that discuss the host as a cause of diversity, and focused on the genetic analysis of T. annulata in Pakistan compared to other endemic sites in Asia, Africa and Europe. 

One more issues is the mingling of local/regional data on parasites diversity. The analyzed hosts in each country used for comparison came from several distinct populations (eg. SW and Central Turkey, Northern and Central Tunis – without exact locations! –, NW, N, NE and SW Oman. While in the case of Oman, the authors treat distinctly two populations (N vs. South) in the case of the other two countries no distinction is made. This approach may introduce a significant geographical bias (the only measure of ‘host diversity’ in the case of this MS is geographical distance between host populations), as for example Aydin (W TR) is chiefly at the same distance to Tunis (TS) as N Oman to Punjab, although genetic distances between these two pairs differ with a magnitude. Moreover, the geographic distance between N and S Oman is small, still the genetic divergence is high. Thus, just geographic distance between host populations per se is useless for such comparisons.

Response: As suggested we combined the North and South Oman populations  of T. anulata in one population, and redid the analysis of diversity (heterozygosity) and population divergence using FST.

The third important issue is the way authors link host-population connectivity to geographical distance. This is entirely wrong, as nowadays genetic diversity in livestock (and especially in commercially bred, globally distributed races in cows) is not maintained exclusively by livestock exchanges as in the past, but by using seminal fluids of selected bulls. And most breeders can and do select from the widest palette of genetic strains, without any correspondence to geographical location… Thus, in this concrete case the first sentence of the abstract is totally erroneous.

Response: As mentioned above we agree with the reviewer on the role of the host. We have edited the Abstract and other sections in the manuscript accordingly.

And a comment on the scope of the study: How will these data improve T. annulata control? The authors need to explain this either in the introduction or the discussion.

Response:  The above point has been addressed above in response to point 5 of  reviewer 2.

 There are several minor issues also:

 1. Language and grammar. Especially the Abstract and several paragraphs of the Discussion are hard to understand and bear many punctuation and grammatical errors.

Response: As mentioned above (response to reviewer 2 point 1), the manuscript was edited for English language, grammar, spelling and consistency of style throughout

 2.L33-34 Abstract/Conclusions: There is no such thing as ‘global endemic site’! It is either global or endemic… Moreover the whole sentence is meaningless.

Response: We thank the review, the sentence has been edited

 3.L71-72 Ticks does not migrate, however they are transported. By the livestock they’re attached and with migratory birds. This later is much more important and at least a magnitude larger, as involves millions of birds transiting this region. So, Hyalomma ticks definitely move between the mentioned areas independent of cattle trade.

Response: We thank the reviewer for this interesting point, Hyalomma anatolicum is the main tick species in the area. However, we do not have information on the parasite in infected ticks. In small servey in Oman we detected Theileria lestoqursdi in infected ticks and found that the infected ticks can carry diverse parasite strains. We added an additional sense to highlight the dispersal of ticks over a long distance via migratory birds (lines 193-194).  

Round 2

Reviewer 2 Report

Thank you for addressing my concerns.

This manuscript is a resubmission of an earlier submission. The following is a list of the peer review reports and author responses from that submission.

Round 1

Reviewer 1 Report

This MS is a short note on genetic identity and variation of Theileria annulata identified in bovine blood in Pakistan in comparison to similar data from Oman, Tunisia and Turkey. The intention of the authors was to assess genetic similarity of T. annulata strains and compare these among geographically distant populations (eg. Pakistan to Turkey, Oman and Tunisia). While the idea itself is interesting, the authors oversimplified the study, with concluding that genetic distances are (1) in correlation with geographical distances, (2) this is caused by genetic make-up of primary hosts (cows). There are several problems with the authors’ approach, making the conclusions of the MS highly questionable.

First of all, the main hypothesis: that host populations and their movements are solely the cause of diversity in T. annulata genetic stocks and these should show a linear correlation with the geographic distances of their locations. While movements of cows (due to international trade primarily) are high, with intermixing at important trade-hubs, this is not the sole cause of genetic diversity in their respective pathogens. It is highly unlikely that Friesian stocks of Pakistani cows introduced the T. annulata genetic lineages to Pakistan (or any other site, by the way). Theileria annulata is vectored by ticks (primarily Hyalomma spp., but not exclusively). There are at least 10 different Hyalomma spp. in Pakistan and similar numbers are present in Turkey, Tunisia and Oman, too. While there is a small overlap between these tick species among the listed countries, each region has several endemics, too (at least compared to the other regions). Thus, one should not exclude the cause of vector linked genetic diversity in case of so many different, locally present vectors. Another logical fallacy associated to this hypothesis is the fact that the authors suggests that bird-flight distances between different sites are corresponding to exact trade-routes of livestock trade. While geographic distances may be correlative in case of transhumance, this is definitely not the case when sea fare is involved (eg. between these countries) and trading hubs are introduced (as in real-world situation).

One more issues is the mingling of local/regional data on parasites diversity. The analyzed hosts in each country used for comparison came from several distinct populations (eg. SW and Central Turkey, Northern and Central Tunis – without exact locations! –, NW, N, NE and SW Oman. While in the case of Oman, the authors treat distinctly two populations (N vs. South) in the case of the other two countries no distinction is made. This approach may introduce a significant geographical bias (the only measure of ‘host diversity’ in the case of this MS is geographical distance between host populations), as for example Aydin (W TR) is chiefly at the same distance to Tunis (TS) as N Oman to Punjab, although genetic distances between these two pairs differ with a magnitude. Moreover, the geographic distance between N and S Oman is small, still the genetic divergence is high. Thus, just geographic distance between host populations per se is useless for such comparisons.

And the third important issue is the way authors link host-population connectivity to geographical distance. This is entirely wrong, as nowadays genetic diversity in livestock (and especially in commercially bred, globally distributed races in cows) is not maintained exclusively by livestock exchanges as in the past, but by using seminal fluids of selected bulls. And most breeders can and do select from the widest palette of genetic strains, without any correspondence to geographical location… Thus, in this concrete case the first sentence of the abstract is totally erroneous.

The methods, results and discussion which have no relevance to the main topic: the haematological indices. Please eliminate the passages in the Results (L 78-87 and Table 1), in the Methods (L206-211) and in the Discussion (144-151).

There are several minor issues also:

  1. In the title: please correct T. annulata.
  2. Language and grammar. Especially the Abstract and the several paragraphs of the Discussion are hard to understand and bear many punctuation and grammatical errors.
  3. Reference 24 and 25 are the same.

Reviewer 2 Report

This study uses mini and micro satellite markers to compare genetic diversity of T. annulata parasites in cattle in Pakistan.  The authors conclude that parasites in different areas develop a distinct pattern of genetic variation that differentiates them from parasites of the same species in a geographically different area.  

I fail to understand why this study is unique, or what it's adds to the field.  How will these data improve T. annulata control? The authors need to add discussion of this in the introduction or discussion.

The role of the pathogen/tick interface in genetic diversity is largely ignored, but is quite important, given that sexual reproduction of the pathogen occurs in the tick.  Please add discussion of tick-related variables.

The authors compare their data to other datasets from T. annulata parasites in other areas.  Are the methods used to assess diversity the same? If not, these comparisons are likely moot.  

Extensive editing is required - "annulata" is misspelled in the title

Reviewer 3 Report

The authors performed a set of analyzes to determine genetic diversity and assess genetic differentiation of T. annulata populations from Pakistan compared to populations from Oman, Tunisia, and Turkey. However, the sample panel used by the authors is not representative of the Pakistani bovine population, which considerably weakens the conclusions obtained by the authors. It is not possible to extrapolate all the results found by the authors in the cattle population from Layyah District of Southern Punjab to the cattle population in Pakistan.

The genetic diversity of Theileria annulata can be considerably influenced by the presence of different species of ticks that parasitize cattle in the studied regions. This aspect was poorly addressed by the authors. Authors should assess whether there was a genetic divergence of Theileria annulata as a function of the tick species present in cattle. This is a confounding factor with the hypothesis that there are different populations of Theileria annulata due to the geographical separation of countries since tick migration occurs with trade or movement of infested cattle between countries.

Other considerations can be found in the attached file.
